# Influence of Hospital Volume of Procedures by Year on the Risk of Revision of Total Hip and Knee Arthroplasties: A Propensity Score-Matched Cohort Study

**DOI:** 10.3390/jcm8050670

**Published:** 2019-05-13

**Authors:** Jorge Arias-de la Torre, Miquel Pons-Cabrafiga, Jose M. Valderas, Jonathan P Evans, Vicente Martín, Antonio J Molina, Francesc Pallisó, Kayla Smith, Olga Martinez, Mireia Espallargues

**Affiliations:** 1Agency for Heath Quality and Assessment of Catalonia (AQuAS), 08005 Barcelona, Spain; ksmith@gencat.cat (K.S.); omartinezcruz@gencat.cat (O.M.); mespallargues@gencat.cat (M.E.); 2CIBER Epidemiology and Public Health (CIBERESP), 28029 Madrid, Spain; vicente.martin@unileon.es; 3Institute of Biomedicine (IBIOMED), University of León, León 24071, Spain; ajmolt@unileon.es; 4Sant Rafael University Hospital, 08035 Barcelona, Spain; 23655mpc@gmail.com; 5Health Services and Policy Research Group, University of Exeter Medical School, Exeter EX1 2LU, UK; J.M.Valderas@exeter.ac.uk (J.M.V.); J.Evans3@exeter.ac.uk (J.P.E.); 6Royal Devon and Exeter NHS Foundation Trust, Exeter EX2 5DW, UK; 7Santa María University Hospital, 25198 Lleida, Spain; fpalliso@gss.scs.es; 8Health Services Research on Chronic Patients Network (REDISSEC), 28029 Madrid, Spain

**Keywords:** total hip arthroplasty, total knee arthroplasty, hospital volume, revision, register studies

## Abstract

The volume of total hip (THA) and knee arthroplasties (TKA) performed in a hospital per year could be an influential factor on the revision of these procedures. The aims of this study were: To obtain comparable cohorts in higher- and lower-volume hospitals; and to assess the association between the hospital volume and the incidence of revision. Data from patients undergoing THA and TKA caused by osteoarthritis and recorded in the Catalan Arthroplasty Register (RACat) between January 2005 and December 2016 were used. The main explanatory variable was hospital volume by year (higher/lower). The cut-off point was fixed, based on previous research, at 50 THA and 125 TKA procedures/year. To obtain comparable populations, a propensity-score matching method (1:1) was used. Patient characteristics prior to and after matching were compared. To assess differences by volume, subhazard ratios (SHRs) from competing risks models were obtained. After matching, 13,772 THA and 36,316 TKA patients remained in the study. Prior to matching, in both joints, significant differences in all confounders were observed between volume groups. After matching, none of them remained significant. Both in THA and TKA, a higher risk of revision in higher-volume hospitals was observed (THA SHR: 1.25, 95%CI: 1.02–1.53; and TKA SHR: 1.29, 95%CI: 1.16–1.44). Unlike other contexts, currently in Catalonia, higher-volume hospitals have a greater risk of revision than lower-volume hospitals. Further research could be valuable to define context-dependent measures to reduce the incidence of revision.

## 1. Introduction

In recent years, several studies have highlighted some specific risk factors for revision surgery in total hip and knee arthroplasties (THA and TKA), including patient factors like sex, age or comorbidity; intervention factors like type of prosthesis or type of fixation; and hospital factors like the circumstances of admission or the number of procedures performed each year [1,2]. The relationship between some of these variables and the risk of revision is clear and consistently found in gender, age and the year the primary procedure was performed [3,4,5]. Nevertheless, the relationship with some hospital and healthcare related variables remains unclear. Due to the possible relationship to the THA and TKA revision, the number of procedures performed per year by the hospital or hospital volume by year has been pointed out [6,7].

Previous studies have shown varying results when examining the relationship between the hospital volume by year and the risk of revision but after focusing on these results, the evidence found is inconsistent [6,7,8,9,10,11,12,13,14,15]. While some of them have not found any association between the hospital volume and the risk of revision [9,10,11], other studies have suggested a possible higher risk of revision for primary prostheses in lower volume hospitals over higher volume, both at medium- and long-term follow-up [6,7,12,13,14]. Differences in the results previously found might be related to the context of the study. As mentioned in prior research, an explanation for the difference in results could be the possible case-mix between volume groups [8,16], meaning the characteristics of patients undergoing an arthroplasty could vary between hospital volume groups. Therefore, it could be advantageous to take this possibility into account to obtain a more accurate representation of how the volume by year could be related to the incidence of revision in THA and TKA.

Regarding the methodology used in the aforementioned studies, most of them were performed without matching cohorts from different volume groups. Due to differences in the characteristics of patients between volume groups, the conclusions drawn from these cohorts might not be comparable. As such, using propensity scores in the last few years to match different populations has become popular [16,17]. These scores might allow comparable cohorts to be acquired and could reduce the bias when comparing the results between them.

Therefore, the aims of this study were: (1) To obtain comparable cohorts in higher and lower volume hospitals and (2) to assess the association between the THA and TKA volume by year and the incidence of revision at one, five and 10 years in Catalonia.

## 2. Experimental Section

### 2.1. Design and Study Population

A prospective observational design based on data from the Catalan Arthroplasty Register (RACat) and the Minimum Basic Dataset at hospital discharge (MBD-HD) was drawn up. The RACat is a population-based arthroplasty register that has collected data about hip and knee arthroplasties performed in Catalonia since 2005. The RACat includes 53 out of 61 public hospitals in Catalonia with approximately 830 surgeons performing hip and knee arthroplasties. The completeness of the RACat is about 90% for primary arthroplasties and about 70% for revision procedures [18]. The MBD-HD is a mandatory, payment-related health database that includes information about the patient and the surgical procedure, like diagnosis (including ICD-9-CM codes) and the reason for intervention and revision. The MBD-HD dataset is merged with the RACat dataset using a common identification number as well as other variables like surgery date and hospital admission and discharge dates. Furthermore, and given its mandatory nature, the MBD-HD is used as a standard to calculate the completeness of the RACat.

The study population included all patients undergoing a THA (conventional) or a TKA (including cruciate retaining and posterior stabilized) caused by osteoarthritis and recorded in the Catalan Arthroplasty Register (RACat) between January 2005 and December 2016. In total, the present study includes 18,283 patients undergoing THA and 45,553 TKA. After matching, a total of 13,772 THA from 46 hospitals (Figure 1) and 36,316 TKA from 49 hospitals (Figure 2) remained in the study.

### 2.2. Study Variables

The main outcome considered in this study was the revision of the primary arthroplasty. A revision procedure is defined within the RACat, both for THA and TKA, as any procedure involving removal, exchange or addition of any implant component.

The volume of THA and TKA procedures performed by year in the hospitals participating in the RACat during the study period was taken as the main exploratory factor. The hospital volume was considered as a dichotomous variable (higher/lower), based both on the cut-off values fixed on previous research on this topic [6,7,12], and on the number of THA and TKA procedures performed by year in the Catalan hospitals. For THA, hospitals performing fewer than 50 procedures per year were considered low-volume hospitals. For TKA, hospitals performing fewer than 125 procedures per year were considered low-volume hospitals. A hospital could provide cases to either the high- or low-volume group, depending on the number of procedures performed in a specific year.

The following confounders were considered: Sex (men and women), age (<65, 65–74, 75–84, and ≥85 years old), comorbidity on the Elixhauser index (no comorbidity, one or two comorbidities, and three or more comorbidities), year of intervention (2005/2007, 2008/2010, 2011/2013, and 2014/2016) and type of fixation (cemented, cementless, hybrid, reverse hybrid, and not specified).

### 2.3. Data Analysis

Descriptive analyses of THA and TKA populations pre- and post-matching were performed, and differences were tested using Chi-Square tests. To select comparable populations, a propensity score matching method 1:1 was used, considering hospital volume as the exposure and sex, age, comorbidity, year of intervention and type of fixation as confounders. The incidence of revision at one, five and 10 years was obtained using the Kaplan–Meier method taking the competing risk of death into account. The incidence of revision was estimated by calculating t S(t−1) × h’(t), where S(t−1) is the Kaplan–Meier estimate of the overall survival function and h’(t) is the cause-specific hazard at time t. In addition, to evaluate differences in risk of revision, competing risks models were fitted, considering death as the competing event and hospital volume as the main exposure. From these models, subhazard ratios (SHRs) for revision and their 95% confidence intervals (95% CI) were obtained. All models and all analyses were stratified by the joint operated on. The significance level was fixed at α = 0.05 and all analyses were performed using Stata version 14 [19].

## 3. Results

In total, 11,132 patients undergoing THA in 28 higher-volume hospitals and 7151 from 46 lower-volume hospitals were included (Table 1). In terms of TKA, a total of 24,683 patients from 30 higher-volume hospitals and 20,870 from 49 lower-volume hospitals were included (Table 2). After matching, 6886 patients undergoing THA and 18,158 undergoing TKA for each volume group remained in the study. Prior to matching, both for THA and TKA, statistically significant differences (*p* < 0.05) in all confounders (sex, age, comorbidity on the Elixhauser index, year of intervention and type of fixation) between higher- and lower-volume groups were observed. After matching, none of these differences remained significant.

In THA, the most frequent causes of revision were aseptic loosening (*n* = 70, 33.7%), infection (*n* = 48, 23.1%), dislocation (*n* = 33, 15.9%) and mechanical complications (*n* = 30, 14.4%) for the higher-volume group, and aseptic loosening (*n* = 48, 28.4%), dislocation (*n* = 46, 27.2%), mechanical complications (*n* = 25, 14.8%) and infection (*n* = 23, 13.6%) for the lower-volume group. In addition, Figure 3 shows the cumulative incidence of revision in THA by the volume group and the magnitude of the differences in the risk of revision between volume groups in THA. The cumulative incidence of revision was 1.31% (95% CI: 1.06–1.61), 2.85% (95% CI: 2.45–3.31) and 4.61% (95% CI: 3.88–5.43) at one, five and 10 years’ follow-up in patients operated on in the higher-volume group hospitals and 0.97% (95% CI: 0.76–1.23), 2.32% (95% CI: 1.95–2.74) and 3.63% (95% CI: 3.06–4.27), respectively in patients from lower-volume hospitals. Taking the differences in the risk of revision into account (Table 3), a higher risk of revision was observed in higher-volume hospitals than in lower-volume hospitals (SHR: 1.25, 95% CI: 1.02–1.53).

Regarding TKA, the most frequent causes of revision were mechanical complications (*n* = 254, 35.7%), aseptic loosening (*n* = 233, 32.7%), infection (*n* = 152, 21.4%) and dislocation (*n* =16, 2.3%) for the higher-volume group, and aseptic loosening (*n* = 252, 44.8%), mechanical complications (*n* = 137, 24.4%), infection (*n* = 88, 15.7%) and dislocation (*n* = 14, 2.5%) for the lower-volume group. Additionally, the cumulative incidence of revision (Figure 4) was 1.15% (95% CI: 1.00–1.32) at one year, 4.17% (95% CI: 3.85–4.50) at five years’ and 5.8% (95% CI: 5.33–6.39) at 10 years’ follow-up for the higher-volume group, and 0.64% (95% CI: 0.53–0.77), 3.24% (95% CI: 2.96–3.53) and 4.7% (95% CI: 4.21–5.13) for the lower-volume group. After examining the differences in risk of revision in TKA (Table 4), a significant higher risk of revision was observed when the volume by year was higher than when it was lower (SHR: 1.29, 95% CI: 1.16–1.44).

## 4. Discussion

Our study overcomes a few methodological limitations of previous research and shows that, nowadays in Catalonia, the volume of THA and TKA procedures performed in a hospital per year could be related to the incidence of revision. Contrary to previous research focused on other contexts carried out using unmatched cohorts [6,7,12,13,14], higher-volume hospitals could have a higher risk of revision in their primary prostheses than lower-volume centers in Catalonia.

Some studies that have analyzed the relationship between the volume of THA and TKA procedures by year and the risk of revision, have not found any relationship between them [9,11]. Nevertheless, recent research carried out with large samples at the national level [6,7,12], both for THA and TKA, have reported results that suggest a higher risk of revision in patients operated on in lower-volume hospitals than in higher-volume hospitals. In this sense, both the study performed with more than 40,000 THA, based on the Nordic Arthroplasty Register Association database [7], and a study with more than 25,000 TKA from Norway [12], suggest that lower-volume hospitals might have an increased long-term risk of revision. Our results were contradictory to this hypothesis, showing an opposite association between the annual hospital THA volume and long-term risk of revision.

As far as we know, only one previous study has found a slightly higher risk of revision in higher-volume hospitals [8]. While the authors of that study hypothesized that their results could be caused by a possible case-mix between volume groups, our study has similar results after overcoming this issue by using propensity-score matched cohorts [16,17]. These results, at least in Catalonia, could be partially explained by factors related to the healthcare organization. Higher-volume hospitals are usually centers that have residents, students in training and more healthcare personnel than lower-volume hospitals. That said, in higher-volume hospitals it is usually much more difficult to deal with environmental factors that might be a potential cause of infection, as was suggested by our results about the causes of revision by the volume group. Additionally, patients from higher-volume hospitals might have been operated on by a surgical trainee, a factor that could be related to higher revision rates due to the learning curve [8,20,21]. In the lower-volume hospitals in Catalonia, usually only one surgeon performs each type of arthroplasty. Due to this, despite the lower volume by hospital, the volume by surgeon could be high, a factor that has been related to lower revision rates [13,14,22]. Further research using matched cohorts from different healthcare contexts, e.g., using data from different countries’ or cultures’ registers, using different cut-off values to divide between higher- and lower-volume groups, and accounting for surgeon and hospital-environment related variables, could help improve our understanding of how volume by year is related to the incidence of revision. Depending on the context, this knowledge might help define specific healthcare organizational measures in an attempt to reduce the burden of revision in THA and TKA.

There are some limitations in this study that need to be discussed. First, the completeness of primary THA in the RACat is not 100%, especially during the first years of its implementation, and it is possible that not all primary and revision procedures were reported in those years. However, no disparity in collecting revisions between high- and low-volume hospitals was observed [18]. Furthermore, the number of prostheses with a follow-up of 10 years is very low but this trend may change in the future. Regarding the specific prosthesis model or brand used in the different hospitals, due to limitations in the robustness and validity of the estimations related to differences between hospitals in their use and to the small sample size of some of these models, this potential confounder was not included. Future research with larger sample sizes accounting for this factor could be valuable in obtaining a more precise representation of how the volume by year affects the outcomes in THA and TKA. Another limitation is the fixed cut-off point to differentiate higher- and lower-volume groups. This variable could be taken as a continuous variable or changed to include a larger number of categories based on different cut-off values. Despite this, these cut-off points were based on those used in previous research with similar aims [6,7,12] and dichotomized due to the low number of patients in some categories after we tested it as a five- and three-category variable. Furthermore, the dichotomization allows us to match the higher- and lower-volume cohorts, making them comparable within the confounders included. In addition, the crossover between hospital volume groups was not taken into account. Nevertheless, as observed in hospitals participating in the RACat [18], this change of the volume group was, in most cases, an increasing or decreasing trend of hospital activity. Finally, due to the data captured in the RACat, it is important to indicate that we studied the hospital volume and not surgeon volume, which may influence our estimates. Furthermore, there is a lack of surgeon-related data and the number of surgeons by hospital is unknown, which are factors that could have a mediating effect between the volume and the risk of revision. In addition, we only considered primary procedure revisions, not other outcomes like patient reported outcome measures (PROMs). Nevertheless, it is important to mention that these limitations could be overcome in the upcoming years because the RACat has been mandatory since 2017, and is beginning to capture surgeon data as well as data on other variables and outcomes like PROMs.

In conclusion, our study shows that hospitals performing a higher volume of arthroplasties in Catalonia have an increased risk of revision due to all causes, compared to low-volume hospitals. Further research taking other variables into account and involving registers from different countries could be valuable to better understand this relationship and define specific context-dependent measures to try to reduce the incidence of revision in both THA and TKA.

## Figures and Tables

**Figure 1 jcm-08-00670-f001:**
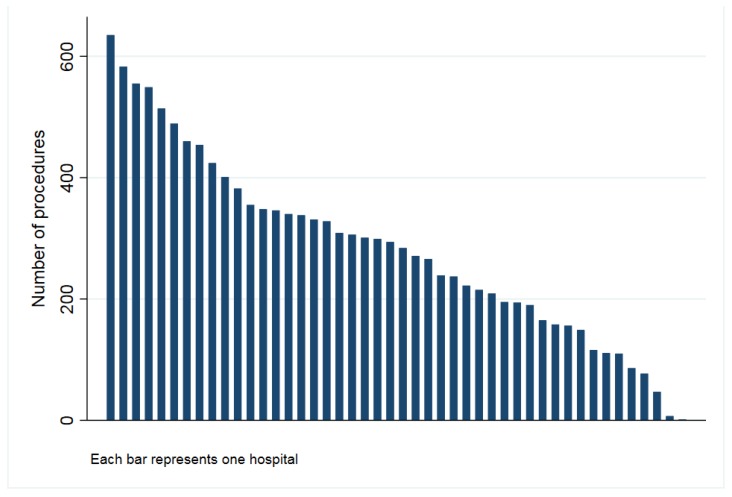
Number of hip procedures by hospital (2005–2016).

**Figure 2 jcm-08-00670-f002:**
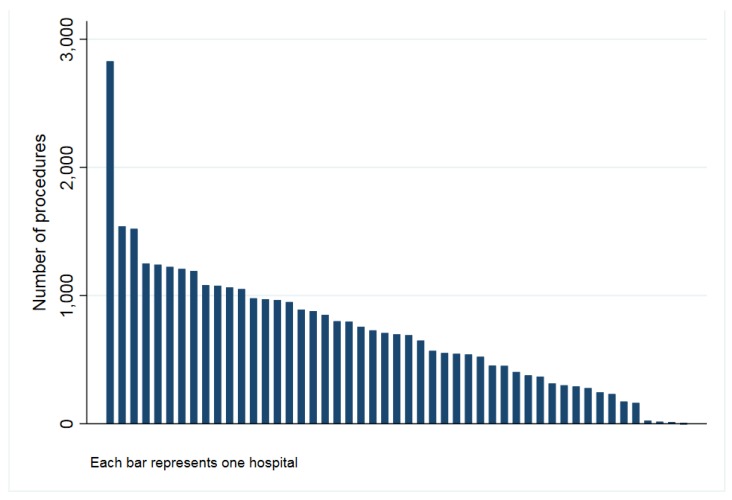
Number of knee procedures by hospital (2005–2016).

**Figure 3 jcm-08-00670-f003:**
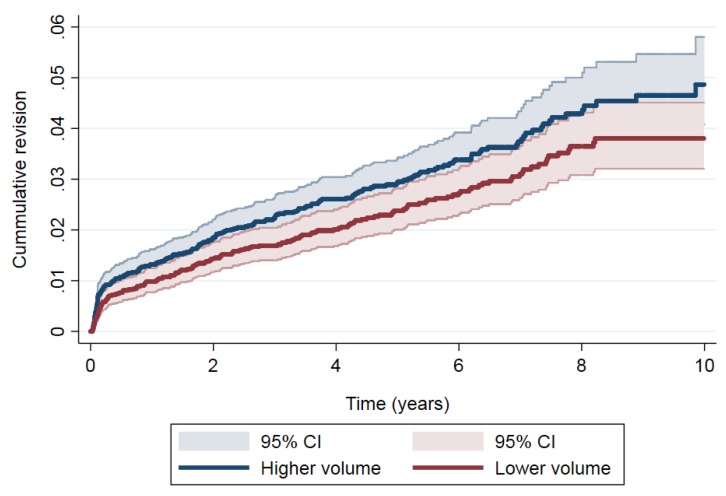
Cumulative revision in total hip arthroplasty (THA).

**Figure 4 jcm-08-00670-f004:**
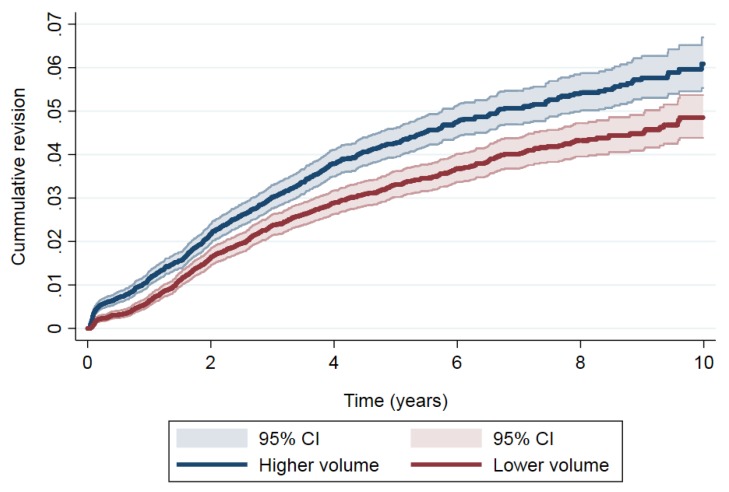
Cumulative revision in total knee arthroplasty (TKA).

**Table 1 jcm-08-00670-t001:** Characteristics of the total hip arthroplasty study population prior to and after matching. RACat 2005–2016.

	Pre-Matching					Post-Matching				
	Higher Volume (*n* = 11,132)		Lower Volume (*n* = 7151)			Higher Volume (*n* = 6886)		Lower Volume (*n* = 6886)		
	*n*	%	*n*	%	*p*	*n*	%	*n*	%	*p*
**Sex**					0.044					0.495
Men	5,439	48.86	3,603	50.38		3,426	49.75	3,466	50.33	
Women	5,693	51.14	3,548	49.62		3,460	50.25	3,420	49.67	
**Age**					0.015					0.416
<65	3,317	29.80	2,047	28.63		2,014	29.25	2,001	29.06	
65–74	3,822	34.33	2,464	34.46		2,331	33.85	2,372	34.45	
75–84	3,562	32.00	2,408	33.67		2,287	33.21	2,293	33.30	
≥85	431	3.87	232	3.24		254	3.69	220	3.19	
**Comorbidity (Elixhauser)**					0.020					0.821
no comorbidity	4,351	39.09	2,65	37.06		2,565	37.25	2,583	37.51	
1–2 comorbidities	5,8	52.10	3,866	54.06		3,723	54.07	3,690	53.59	
3 or more comorbidities	981	8.81	635	8.88		598	8.68	613	8.90	
**Year of intervention**										0.993
2005/2007	982	8.82	1,116	15.61		875	12.71	868	12.61	
2008/2010	2,999	26.94	2,038	28.50		2,025	29.41	2,035	29.55	
2011/2013	3,871	34.77	2,253	31.51		2,258	32.79	2,248	32.65	
2014/2016	3,280	29.46	1,744	24.39		1,728	25.09	1,735	25.20	
**Type of fixation**					<0.001					0.700
Cemented	906	8.14	719	10.05		666	9.67	636	9.24	
Cementless	7,561	67.92	4,93	68.94		4,762	69.15	4,786	69.50	
Hybrid	2,512	22.57	1,397	19.54		1,388	20.16	1,384	20.10	
Reverse hybrid	106	0.95	93	1.30		70	1.02	80	1.16	
Not specified	47	0.42	12	0.17		.	.	.	.	
**Number of hospitals**	28		46			28		46		
**Median follow-up (IQR)**	4.6 (4.5)					4.8 (4.7)				

Higher volume: Fifty or more procedures per year in total hip arthroplasty and 125 or more procedures per year in total knee arthroplasty; n: number of patients; %: Percentage of patients; *p*: *p* value based on Chi-Square tests; comorbidity (Elixhauser): Number of comorbidities from Elixhauser index.

**Table 2 jcm-08-00670-t002:** Characteristics of the total knee arthroplasty study population prior to and after matching. RACat 2005–2016.

	Pre-Matching					Post-Matching				
	Higher Volume (*n* = 24,683)		Lower Volume (*n* = 20,870)			Higher Volume (*n* = 18,158)		Lower Volume (*n* = 18,158)		
	*n*	%	*n*	%	*p*	*n*	%	*n*	%	*p*
**Sex**					<0.001					0.709
Men	6,971	28.24	6,220	29.80		5,140	28.31	5,108	28.13	
Women	17,712	71.76	14,65	70.20		13,018	71.69	13,050	71.87	
**Age**					<0.001					0.950
<65	4,02	16.29	3,608	17.29		3,117	17.17	3,157	17.39	
65–74	10,747	43.54	9,151	43.85		8,015	44.14	7,976	43.93	
75–84	9,363	37.93	7,767	37.22		6,729	37.06	6,729	37.06	
≥85	553	2.24	344	1.65		297	1.64	296	1.63	
**Comorbidity (Elixhauser)**					<0.001					0.886
no comorbidity	6,633	26.87	6,533	31.30		5,154	28.38	5,115	28.17	
1–2 comorbidities	14,871	60.25	12,01	57.55		10,860	59.81	10,904	60.05	
3 or more comorbidities	3,179	12.88	2,327	11.15		2,144	11.81	2,139	11.78	
**Year of intervention**					<0.001					0.756
2005/2007	2,994	12.13	3,287	15.75		2,328	12.82	2,388	13.15	
2008/2010	6,846	27.74	6,097	29.21		5,493	30.25	5,428	29.89	
2011/2013	7,136	28.91	6,36	30.47		5,711	31.45	5,700	31.39	
2014/2016	7,707	31.22	5,126	24.56		4,626	25.48	4,642	25.56	
**Type of fixation**					<0.001					0.292
Cemented	18,901	76.57	16,769	80.35		15,619	86.02	15,616	86.00	
Cementless	205	0.83	152	0.73		91	0.50	148	0.82	
Hybrid	3,491	14.14	2,545	12.19		2,421	13.33	2,367	13.04	
Reverse hybrid	184	0.75	30	0.14		27	0.15	27	0.15	
Not specified	1,902	7.71	1,374	6.58		.	.	.		
**Number of hospitals**	30		49			30		49		
**Median follow-up (IQR)**	4.8 (5.0)					4.9 (4.7)				

Higher volume: Fifty or more procedures per year in total hip arthroplasty and 125 or more procedures per year in total knee arthroplasty; n: Number of patients; %: Percentage of patients; p: *p* value based on Chi-Square tests; comorbidity (Elixhauser): Number of comorbidities from Elixhauser index.

**Table 3 jcm-08-00670-t003:** Cumulative revision and differences by hospital volume of procedures by year in total hip arthroplasty (THA). RACat 2005–2016.

Time (in Years)	At Risk	Revisions	Cum. Failure (95% CI)	SHR (95% CI)
**Lower volume**				1.00
1	6,258	66	0.97 (0.76–1.23)	
5	3,353	70	2.32 (1.95–2.74)	
10	352	32	3.63 (3.06–4.27)	
**Higher volume**				1.25 (1.02–1.53)
1	6,239	89	1.31 (1.06–1.61)	
5	3,214	82	2.85 (2.45–3.31)	
10	389	36	4.61 (3.88–5.43)	

At risk: Number of primary procedures at the beginning of the period. Revisions: Number of revision procedures during the period; cum. failure: Cumulative failure; (95% CI): 95% confidence interval. SHR: Subhazard ratio for the hospital volume (Lower volume as reference category) adjusted for sex, age, comorbidity (from Elixhauser index), year of intervention and type of fixation.

**Table 4 jcm-08-00670-t004:** Cumulative revision and differences by hospital volume of procedures by year in total knee arthroplasty (TKA). RACat 2005–2016.

Time (in Years)	At Risk	Revisions	Cum. Failure (95% CI)	SHR (95% CI)
**Lower volume**				1.00
1	16,594	112	0.64 (0.53–0.77)	
5	9,064	366	3.24 (2.96–3.53)	
10	1,217	80	4.66 (4.21–5.13)	
**Higher volume**				1.29 (1.16–1.44)
1	16,293	202	1.15 (1.00–1.32)	
5	8,673	422	4.17 (3.85–4.50)	
10	744	89	5.84 (5.33–6.39)	

At risk: Number of primary procedures at the beginning of the period. Revisions: Number of revision procedures during the period; cum. failure: Cumulative failure; (95% CI): 95% confidence interval. SHR: Subhazard ratio for the hospital volume (Lower volume as reference category) adjusted for sex, age, comorbidity (from Elixhauser index), year of intervention and type of fixation.

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
