# Peer review of "Influence of Hospital Volume of Procedures by Year on the Risk of Revision of Total Hip and Knee Arthroplasties: A Propensity Score-Matched Cohort Study"

_jcm, 2019, doi:10.3390/jcm8050670_

Reviewer 1 Report

This is a sound analysis of the impact of volume of TJR to revision rates. The result is interesting as it shows that existing assumptions might be questioned. 

It would be interesting to know the reasons for revision (loosening, dislocation, infection, material failure etc.) within the respective groups. This could further improve the value of the manuscript.

Author Response

Dear reviewer,

We would like to thank you for the effort made and time spent reviewing our article. Our responses to the comments can be seen in the accompanying point-by-point document, and we have edited the main document accordingly.

Please do not hesitate to contact us for any additional information you may need.

Again, thank you very much for your time and consideration and we hope that the changes meet your expectations.

# Reviewer comments

This is a sound analysis of the impact of volume of TJR to revision rates. The result is interesting as it shows that existing assumptions might be questioned. 

It would be interesting to know the reasons for revision (loosening, dislocation, infection, material failure etc.) within the respective groups. This could further improve the value of the manuscript.

RE: We agree with the reviewer’s suggestion and have included the most frequent causes of revision for the different volume groups.

Changes in text:

Please see Methods line 82

Please see Results lines 148 to 155 and 169 to 173.

Please see Discussion lines 210 to 211

Reviewer 2 Report

The authors present a well performed registry study regarding risk of revision according to high and low volume hospitals for total knee arthroplasty and total hip arthroplasty. The methods are sound and the discussion is nicely balanced with limitations properly addressed.

I have a few suggestions to further improve the paper:

1 Number of TKA or THA per hospital

A) It would be nice to have some information on the distribution of procedures TKA and THA for the hospitals (e.g. histogram).

B) Why not use the number of TKA and THA as a continuous variable in the analyses?

C) It would be interesting to see the results with different cut off values e.g. 20 for THA and 50 for TKA are used.

2 Procedures per hospital versus procedures per surgeon

A) Orthopaedic surgeons perform the surgeries, so it makes sense to look at the number of procedures per surgeon instead of hospital. Large hospital may have surgeons performing a low number of procedures each year alongside surgeons performing a high number of procedures each year. Could you perform an analysis for procedures per surgeon?

B) Regarding the number of procedures per hospital. How many surgeons are the in the included  hospitals? This could be a possible confounder.

3 Brand of implant as possible confounder

103: “The following confounders were considered: …”

Type/brand of implant is also a possible confounder. Hence, it is recommended to also correct for  “type/brand of implant” (e.g. nexgen, triathlon etc)  as a confounder, because high volume centres may use different brands of implants than low volume hospitals.

Author Response

Dear reviewer,

We would like to thank you for the effort made and time spent reviewing our article. Our responses to the comments can be seen in the accompanying point-by-point document, and we have edited the main document accordingly.

Please do not hesitate to contact us for any additional information you may need.

Again, thank you very much for your time and consideration and we hope that the changes meet your expectations.

# Reviewer comments

The authors present a well performed registry study regarding risk of revision according to high and low volume hospitals for total knee arthroplasty and total hip arthroplasty. The methods are sound and the discussion is nicely balanced with limitations properly addressed.

I have a few suggestions to further improve the paper:

1 Number of TKA or THA per hospital

A) It would be nice to have some information on the distribution of procedures TKA and THA for the hospitals (e.g. histogram).

RE: We appreciate this comment and have now included two graphs, one for each joint, to show the distribution of the number of procedures included by hospital after matching.

Changes in text:

Please see lines 89 to 91 of Methods

Please see Figure 1 and Figure 2

B) Why not use the number of TKA and THA as a continuous variable in the analyses?

RE: We appreciate this comment and though with the categorization we did lose some information, it might be valuable to define future strategies, healthcare circuits, and specific considerations aimed to reduce the burden of revision in hip and knee arthroplasty. In addition, the dichotomization allows us to match the higher and lower volume cohorts by propensity score, enabling comparison between the confounders included in the score. These aspects were included in the limitations section.

Changes in text:

Please see discussion lines 217 to 218, 234 to 235, and 238 to 239.

C) It would be interesting to see the results with different cut off values e.g. 20 for THA and 50 for TKA are used.

RE: We completely agree with this comment. Given the distribution of patients per year in the hospitals from our region, we decided to dichotomize the volume variable. We would like to point out that the cut-off values were fixed based on previous studies to maximize the comparability with their results. Additionally, as explained in the previous comment, the dichotomization allows us to match the volume cohorts, making them comparable in the confounders included. These aspects as well as a recommendation to take them into account in future studies were added to the text.

Changes in text:

Please see discussion lines 217 to 218, 234 to 235, and 238 to 239.

2 Procedures per hospital versus procedures per surgeon

A) Orthopaedic surgeons perform the surgeries, so it makes sense to look at the number of procedures per surgeon instead of hospital. Large hospital may have surgeons performing a low number of procedures each year alongside surgeons performing a high number of procedures each year. Could you perform an analysis for procedures per surgeon?

RE: We completely agree with the reviewer but unfortunately, our registry did not capture data at the surgeon level until 2017. Regarding the lack of surgeon-related information, we have expanded our explanation in the limitations section.

Changes in text:

Please see lines 244 to 249 of Discussion

B) Regarding the number of procedures per hospital. How many surgeons are the in the included  hospitals? This could be a possible confounder.

RE: We have included an approximation of the number of surgeons performing in Cataluña and information about the possible mediating effect between hospital volume and the risk of revision in the limitations section. 

Changes in text:

Please see lines 80 to 81 of Methods.

Please see lines 244 to 249 of Discussion

3 Brand of implant as possible confounder

103: “The following confounders were considered: …”

Type/brand of implant is also a possible confounder. Hence, it is recommended to also correct for  “type/brand of implant” (e.g. nexgen, triathlon etc)  as a confounder, because high volume centres may use different brands of implants than low volume hospitals.

RE: We completely agree with this comment and the possible confounding role of prosthesis brands. Therefore, due to the limitations related to differences between hospitals in prosthesis brand use and to the small sample size of some of these models, this potential confounder was not included in the propensity score. We would like to explain here that including the prosthesis brand in the matching score would probably yield an extremely small cohort, thus compromising the robustness of the estimates and their validity during analysis. Despite this, we have added it as a limitation of the study and suggested that this factor be included in future research on the topic.

Changes in text:

Please see Discussion lines 228 to 233